## Technical Note:

# New insights into stomatal oxygen transport viewed as a multicomponent diffusion process

Jordi Vilà-Guerau de Arellano<sup>1</sup>, Roderick Dewar<sup>2,3</sup>, Kim A. P. Faassen<sup>1</sup>, Teemu Hölttä<sup>4</sup>, Remco de Kok<sup>1</sup>, Ingrid T. Luijkx<sup>1</sup>, and Timo Vesala<sup>2,4</sup>

**Correspondence:** Jordi Vilà-Guerau de Arellano (jordi.vila@wur.nl)

#### Abstract.

We investigate oxygen  $(O_2)$  transport through stomata, focusing on its interaction with water vapour  $(H_2O)$  flux. The dominant  $H_2O$  flux exerts a drag force on other gases, a well-studied effect in the ternary air–water vapor–carbon dioxide  $(CO_2)$  system but unexplored for  $O_2$  transport. This study aims to: (1) apply the Stefan-Maxwell equations to a quaternary system of  $H_2O$ ,  $O_2$ ,  $CO_2$ , and  $N_2$ ; (2) identify conditions where  $O_2$  transport from stomata to the atmosphere occurs against its mole fraction gradient ('uphill'); and (3) derive an expression linking the  $O_2$  mole fraction in sub-stomatal air spaces  $(x_{oi})$  to that in the atmosphere  $(x_{oa})$  based on atmospheric relative humidity.

Our theoretical results, constrained by typical values from previous flux observations of this quaternary system, reveal distinct transport regimes defined by the mole flux ratio of  $H_2O$  and  $O_2$  ( $F_w/F_o$ ). Uphill  $O_2$  diffusion occurs in the common regime where  $F_w/F_o$ »1, and internal  $O_2$  mole fraction increases towards its atmospheric value as relative humidity tends to 100%. These theoretical results offer a framework for interpreting laboratory and field experiments on stomatal  $O_2$  exchange under stagnant atmospheric or low Reynolds number conditions and can support the development of more physically accurate models of leaf–atmosphere oxygen exchange.

<sup>&</sup>lt;sup>1</sup>Meteorology and Air Quality Section, Wageningen University and Research, The Netherlands

<sup>&</sup>lt;sup>2</sup>Institute for Atmospheric and Earth System Research/Physics, Faculty of Science, University of Helsinki, Finland

<sup>&</sup>lt;sup>3</sup>Division of Plant Sciences, Research School of Biology, The Australian National University, Canberra, Australia

<sup>&</sup>lt;sup>4</sup>Institute for Atmospheric and Earth System Research/Forest Sciences, Faculty of Agriculture and Forestry, University of Helsinki, Finland

#### 1 Introduction

The terrestrial biosphere exchange of water (H<sub>2</sub>O), carbon dioxide (CO<sub>2</sub>) and oxygen (O<sub>2</sub>), plays a crucial role in the global water and carbon cycles. These cycles integrate processes across scales, from the stomatal level to the global atmosphere. Transport between stomata and the atmosphere represents the smallest-scale process governing the exchange of greenhouse gases. When scaled up to leaf and canopy levels, this stomata-atmosphere interaction modifies surface fluxes, which directly influence the diurnal variability of carbon tracers (Faassen et al., 2023) and impacts weather (Jacobs and de Bruin, 1997; Boussetta et al., 2013) and climate-carbon patterns (Bonan et al., 2024). Advancing our understanding of these interactions requires a more integrated approach that bridges ecophysiology and transport dynamics across multiple scales (Vilà-Guerau de Arellano et al., 2023; Miralles et al., 2025). In doing so, we improve the representation of carbon tracers across a wide range of scales.

While the exchange of CO<sub>2</sub> (Goudriaan, 1977; von Caemmerer and Farquhar, 1981) and H<sub>2</sub>O (Jarvis and McNaughton, 1986), typically quantified through molar flux density, has been extensively studied and well-characterized, the transport of O<sub>2</sub> between stomata and the atmosphere has received comparatively little attention. Recent advances now allow biosphere-atmosphere exchange of O<sub>2</sub> to be measured with increasing accuracy (Stephens et al., 2007; Ishidoya et al., 2013; Battle et al., 2019; Faassen et al., 2023), supported by conceptual land-atmosphere modelling studies (Yan et al., 2023; Faassen et al., 2024). This provides a unique opportunity to study the H<sub>2</sub>O-CO<sub>2</sub>-O<sub>2</sub> system through a complete and integrated approach, advancing our understanding of processes related to photosynthesis and the transport of carbon tracers. Notably, O<sub>2</sub> transport exhibits distinct characteristics: O<sub>2</sub> exists at a much higher mole fraction in the atmosphere than either CO<sub>2</sub> or water vapour, yet its flux is of the same order of magnitude as that of CO<sub>2</sub> and typically two to three orders of magnitude smaller than the water vapour flux. As a result, the O<sub>2</sub> flux may be influenced by fluxes of other gases. The significance of this issue can strongly affect the interpretation of O<sub>2</sub> exchange on stomatal level, as was recently highlighted by (Vesala, 2024), whereas on the scale of canopy exchange the O<sub>2</sub>-transport is dominated by turbulent fluxes (Ishidoya et al., 2013; Faassen et al., 2023).

The stomatal–atmosphere exchange is affected by molecular interaction between gases (also referred to as species). This exchange is primarily driven by the dominant effect of water (*i.e.* stomatal transpiration) relative to the individual fluxes of O<sub>2</sub> and CO<sub>2</sub>. Specifically, transpiration at the stomatal level generates drag forces (also referred to as frictional forces) and a bulk displacement of air known as Stefan (or molar) flow. This process leads to gradients between the sub-stomatal cavity and the atmosphere mole fractions, thereby modifying their diffusive transport. Traditionally, the relationship between the flux of a given species and its mole fraction gradient is described by Fick's law, but the additional drag forces associated with multicomponent interactions can lead to the emergence of more complex (non-Fickian) flux–gradient relationships. Accurately interpreting measurements of O<sub>2</sub>, CO<sub>2</sub>, and H<sub>2</sub>O, as well as their representation in models, requires these species–species interactions to be taken into account, e.g. Jarman (1974). In this framework, the Stefan–Maxwell equations provide a more complete description than Fick's law, unifying the effects of molar flow, interspecies drag (friction), and diffusive flux into a single formulation. Fick's law can be deduced from the Stefan–Maxwell equations as an approximation.

To address this in the case of O<sub>2</sub>, we investigate the transport of O<sub>2</sub>, CO<sub>2</sub>, H<sub>2</sub>O, and the inert carrier nitrogen (N<sub>2</sub>) between the leaf intercellular space within the stomata and the atmosphere. Our approach is based on the well-established theory of multicomponent diffusion (Jarman, 1974). Mathematically, the theory is encapsulated in the Stefan–Maxwell equations, which describe the net balance between the thermodynamic force (*i.e.* mole fraction gradient) on a given species that drives its diffusion and the momentum exchange (frictional drag) arising from binary collisions with all the other species, which depends on their relative molar velocities. Previous studies have shown that Stefan flow can play a significant role in CO<sub>2</sub> transport at the stomatal level (Jarman, 1974), and may also be relevant at larger atmospheric scales—from the canopy to the atmospheric boundary layer—as well as for key compounds involved in the photosynthesis cycle, such as O<sub>2</sub> (Kowalski et al., 2021). However, a comprehensive mathematical description of oxygen exchange in relation to water vapour, carbon dioxide, and nitrogen at the stomatal level remains absent.

We adopt a two-step approach. First, we analyze a binary mixture following the method described in Lushnikov et al. (1994) where one species diffuses in a stagnant carrier species. Secondly, we examine a multicomponent (quaternary) system consisting of  $H_2O$ ,  $O_2$ ,  $CO_2$  diffusing in stagnant  $N_2$ , building on previous analyses of ternary systems, such as  $H_2O$ - $CO_2$ -air (Jarman, 1974; von Caemmerer and Farquhar, 1981). Our key objectives are:

- 1. To derive interrelated molar flux density and gradient expressions for O<sub>2</sub>, H<sub>2</sub>O, and CO<sub>2</sub> that account for the net balance between drag and thermodynamic forces.
- 2. To identify general conditions under which the transport of O<sub>2</sub> occurs against its mole fraction gradient, referred here as 'uphill', due to the drag effect of transpiration.
- 3. To derive an expression for the internal stomatal  $O_2$  mole fraction as a function of atmospheric relative humidity.

65

To achieve these objectives, we apply the Stefan-Maxwell equations to the system composed of  $N_2$ - $O_2$ - $H_2O$ - $CO_2$ . We focus on the transport processes between stomata and the atmosphere, without explicitly accounting for the biochemistry of  $O_2$  sources and sinks within the leaf. While our study is based on theoretical calculations, these are constrained by representative measurements of water vapour, carbon dioxide, and oxygen fluxes at the leaf level (Vilà-Guerau de Arellano et al., 2020), which are subsequently scaled down to the stomatal level at which the Stefan-Maxwell equations apply.

The rest of the paper is structured as follows. In section 2 we outline the three key elements of multicomponent diffusion theory that were used in this study: molar balance, decomposition of species transport into molar flow and molecular diffusive flux components, and the Stefan-Maxwell equations relating species mole fraction gradients to species flux densities. In section 3, as a simple reference case, we use this theory to derive expressions for gas transport in a binary system consisting of one tracer species diffusing in stagnant air (e.g. (Jarman, 1974)). In section 4 we extend this theory to a more realistic quaternary system consisting of  $O_2$ ,  $O_2$  and  $O_3$  diffusing in stagnant  $O_3$ . In particular, we derive a simple expression linking the  $O_3$  mole fraction gradient to relative humidity. We then identify the general regime conditions of  $O_3$  transport with respect to the water vapour flux. We discuss the implications of these results in Section 5.

#### 2 Multicomponent diffusion theory

To study the exchange of O<sub>2</sub> between a single stoma and a stagnant atmosphere relating to H<sub>2</sub>O, CO<sub>2</sub> and N<sub>2</sub>, we derive a theory that is applied to gas transport through a single stomatal pore. Our derivations are at the stomatal level, but can be scaled up to the leaf level. When discussing our results numerically using representative leaf-scale flux observations (Vilà-Guerau de Arellano et al., 2020), we convert from stomatal-scale flux densities (mol m<sup>-2</sup> pore s<sup>-1</sup>) to leaf-scale flux densities (mol m<sup>-2</sup> leaf s<sup>-1</sup>). In short, by multiplying the stomatal-scale flux density by the cross-sectional area of a pore (m<sup>2</sup> pore), we obtain the flux through a single stoma. Multiplying this value by the stomatal density (number of stomata per m<sup>2</sup> of leaf surface) then yields the flux density at the leaf scale.

Our key conclusions do not depend on this conversion factor, since this it cancels out when scaling (or downscaling) from the stomata to the atmosphere. For simplicity we consider one-dimensional gas transport through a stomatal pore connecting the stomatal interior to the external atmosphere. Appendix B describes the geometrical assumptions underlying this simplification. The theory presented in the next section can readily be generalized to three spatial dimensions. Finally, for clarity, Table A1 in Appendix A, provides symbol definitions, units, and representative parameter values, including typical values for mole fractions, prescribed molar flux densities based on observations, and molecular diffusivity coefficients.

#### 2.1 Molar balance

Molar balance of a tracer provides the overall context for the description of multicomponent diffusion. The molar balance equation for  $c_{\alpha}$ , the concentration of chemical species  $\alpha$  (units mol m<sup>-3</sup>), reads

$$\frac{\partial c_{\alpha}}{\partial t} = -\frac{\partial F_{\alpha}}{\partial l} = R_{\alpha},\tag{1}$$

where  $F_{\alpha}$  (mol m<sup>-2</sup> s<sup>-1</sup>) and  $R_{\alpha}$  are, respectively, the stomatal-scale molar flux density and the net production rate (*i.e.*) sources - sinks) of species  $\alpha$ , t is time and t is the distance along the pore (see Fig. B1). For t species, the total mole fraction of the mixture is

$$100 \quad c_t = \sum_{\alpha=1}^n c_\alpha. \tag{2}$$

and Eq. 1 can then be re-expressed in terms of the mole fraction  $x_{\alpha} = c_{\alpha}/c_{t}$  (t is here the total amount of species), as

$$\frac{\partial x_{\alpha}}{\partial t} = -\frac{1}{c_{t}} \frac{\partial F_{\alpha}}{\partial l} + r_{\alpha},\tag{3}$$

where  $r_{\alpha} = R_{\alpha}/c_{t}$ .

#### 2.2 Decomposition of fluxes into molar flow and diffusive flux

For each chemical species  $\alpha$ , we can decompose its molar flux density  $F_{\alpha}$  into a component related to the average flow of the mixture as a whole (molar flow, also called Stefan flow) and a diffusive component related to its transport relative to this average (diffusive flux). We first note that  $F_{\alpha}$  is related to the species velocity  $u_{\alpha}$  (relative to the stationary laboratory frame) by

$$F_{\alpha} = c_{\alpha} u_{\alpha} \tag{4}$$

The mole-average velocity of the mixture (u) is defined as the sum of the species velocities weighted by their respective mole fractions:

$$u = \sum_{\alpha=1}^{n} x_{\alpha} u_{\alpha},\tag{5}$$

By substituting  $x_{\alpha} = c_{\alpha}/c_t$  into Eq. 5 and using Eq. 4, u can be expressed in terms of the total molar flux density of the mixture ( $F_t$ ):

$$115 \quad u = \frac{F_t}{c_t},\tag{6}$$

where

$$F_t = \sum_{\alpha=1}^n F_\alpha,\tag{7}$$

We can then decompose  $F_{\alpha}$  (Eq. 4) into the sum of a Stefan flow defined by

$$s_{\alpha} = c_{\alpha} u,$$
 (8)

120 and a diffusive flux defined by

$$f_{\alpha} = c_{\alpha}(u_{\alpha} - u). \tag{9}$$

Using Eqs 4 and 9, we can find an expression of the flux  $F_{\alpha}$  related to the molar flow and the diffusive flux:

$$F_{\alpha} = c_{\alpha} u_{\alpha} = s_{\alpha} + f_{\alpha}. \tag{10}$$

From these relationships we note that the total diffusive flux of the mixture is zero, *i.e.*.  $\sum_{\alpha=1}^{n} f_{\alpha} = 0$ .

Whereas Eqs. 8 and 9 define Stefan flow and diffusion in terms of species velocities, we can use Eqs. 6 and 10 to re-express these components in terms of the species molar flux densities, giving

$$s_{\alpha} = x_{\alpha} F_t \tag{11}$$

and

140

$$f_{\alpha} = F_{\alpha} - x_{\alpha} F_t. \tag{12}$$

From Eq. 11 we note that the Stefan flow of a given species is proportional to its mole fraction so that, for a given total flux density  $F_t$ , we expect Stefan flow to be much larger for  $O_2$  ( $x_o$ =0.21) than for  $H_2O$  ( $x_w=10^{-2}$ ) or  $CO_2$  ( $x_c=4\times10^{-4}$ ) (see Table A for additional values). Using the decomposition of Eqs. 8-10, we can then re-express molar balance (Eq. 3) as

$$\frac{\partial x_{\alpha}}{\partial t} + \frac{\partial (x_{\alpha}u)}{\partial l} = -\frac{1}{c} \frac{\partial f_{\alpha}}{\partial l} + r_{\alpha}. \tag{13}$$

where the first and second terms on the right-hand side represent the contributions to mass transport from Stefan flow and diffusion, respectively.

To complete our analysis of the molar  $O_2$ -equation, and focusing on the molar balance for oxygen, Appendix D shows Eq. 13 in a dimensionless form. The analysis shows that the diffusive term can be ignored at atmospheric scales — due to the dominance of turbulence with respect the molecular diffusion transport — . However, at the stomatal scale, characterized by low Reynolds numbers (rougly speaking maximum values of  $10^4$ , the Stefan flow and the diffusive terms needs to be included. It is at this stomatal scale, and in order to solve Eq. 13 for the mole fractions  $x_{\alpha}$ , we need to relate the Stefan flow and diffusion terms to the species mole fractions. This linkage is supplied by the Stefan-Maxwell equations, and it is at this stage that the need for a multicomponent approach becomes apparent.

#### 2.3 Stefan-Maxwell equations

Stefan-Maxwell equations provide a framework to study the dependence of the flux as a function of the gradients of other of all other species. For a multicomponent system consisting of n chemical species, the relationship between the one-dimensional mole fraction gradient of species  $\alpha$  and the species flux densities is given by the Stefan-Maxwell equation (Curtiss and Hirschfelder, 1949; Kalkkinen et al., 1991; Bird et al., 2007):

$$\frac{\partial x_{\alpha}}{\partial l} = \sum_{\beta=1,\beta\neq\alpha}^{n} \frac{x_{\alpha} F_{\beta} - x_{\beta} F_{\alpha}}{c D_{\alpha\beta}} = \sum_{\beta=1,\beta\neq\alpha}^{n} \frac{x_{\alpha} f_{\beta} - x_{\beta} f_{\alpha}}{c D_{\alpha\beta}} = \sum_{\beta=1,\beta\neq\alpha}^{n} \frac{x_{\alpha} x_{\beta} (u_{\beta} - u_{\alpha})}{D_{\alpha\beta}},$$
(14)

where  $D_{\alpha\beta}$  is the binary diffusivity of species  $\alpha$  and  $\beta$ , and the summations are over all species  $\beta$  other than species  $\alpha$ . The second equality follows (i) substituting  $F_{\alpha}$  and  $F_{\beta}$  in the numerator using Eq. 12 and (ii) cancellation of the Stefan flow components ( $x_{\alpha}F_{t}$  and  $x_{\beta}F_{t}$ ). The last and third equality follows substituting  $F_{\alpha}$  and  $F_{\beta}$  using Eq. 4.

Physically, Eq. 14 describes the net balance of forces on species  $\alpha$ . The left-hand side is the thermodynamic driving force for diffusion of species  $\alpha$  while the right-hand side is the sum of the drag (frictional) forces due to molecular collisions between species  $\alpha$  and all the other species. The drag force on species  $\alpha$  due to collisions with species  $\beta$  is proportional to the product of their mole fractions  $(x_{\alpha}x_{\beta})$  and their velocity difference  $(u_{\beta}-u_{\alpha})$ . The inverse of the binary diffusivity  $(1/D_{\alpha\beta})$  then represents an effective drag coefficient. A useful feature of the Stefan-Maxwell equations is that the binary diffusivities are virtually independent of species mole fractions (Bird et al., 2007). As shown below (sections 3 and 4), Eq. 14 can be inverted to express the molar flux density of species  $\alpha$  ( $F_{\alpha}$ ) in terms of the spatial gradients of all other species in the system. This facilitates the comparison with Fick's law. It also shows the implications of solving the molar balance equation (Eq. 13) using a multicomponent approach that accounts for the interactions between the different species.

#### 3 Binary mixture: one species diffusing in stagnant air

We first apply Eq. 14 to a binary system (n=2) consisting of one tracer species (e.g.  $\alpha = H_2O$ ,  $O_2$  or  $CO_2$ ) diffusing in stagnant air. Here we define air composed solely by nitrogen and indicated by the subscript a. This provides a simplified but useful reference case against which the more realistic quaternary system will be compared (section 4). In this case there is only one term on the right-hand side of Eq. 14 ( $\beta = \alpha$ ). Substituting  $F_a = 0$  (stagnant air, i.e.,  $u_a = 0$ , where a means air) and  $x_a = 1 - x_\alpha$ , then solving for  $F_\alpha$  gives:

$$F_{\alpha} = -\frac{c_t D_{\alpha a}}{1 - x_{\alpha}} \left( \frac{\partial x_{\alpha}}{\partial l} \right), \tag{15}$$

where  $D_{\alpha a}$  is the binary diffusivity of species  $\alpha$  in air. The total molar flux density of the binary mixture is simply  $F_t = F_{\alpha} + F_a = F_{\alpha}$  (since  $F_a = 0$ ), so from Eqs. 11, 12 and 15 the Stefan flow and diffusive components of  $F_{\alpha}$  are given by

170 
$$s_{\alpha} = x_{\alpha} F_{\alpha} = -\frac{x_{\alpha}}{1 - x_{\alpha}} c_t D_{\alpha a} \left( \frac{\partial x_{\alpha}}{\partial l} \right)$$
 (16)

and

160

$$f_{\alpha} = (1 - x_{\alpha})F_{\alpha} = -c_t D_{\alpha a} \left(\frac{\partial x_{\alpha}}{\partial l}\right) \tag{17}$$

Eq. 15 shows that for this simple binary system, the molar flux density of the diffusing species  $(F_{\alpha})$  is proportional to its mole fraction gradient  $(\frac{\partial x_{\alpha}}{\partial l})$ , i.e. a Fickian diffusion law, with an effective diffusion coefficient  $D_{\alpha a}/(1-x_{\alpha})$  that depends on the mole fraction  $x_{\alpha}$ . In section 4, we contrast this result with the four-component system  $(N_2-O_2-H_2O-CO_2)$  for which, in general,

the molar flux densities of water and oxygen  $(F_w \text{ and } F_o)$  each depend on both mole fraction gradients  $(\left(\frac{\partial x_w}{\partial l}\right))$  and  $(\left(\frac{\partial x_o}{\partial l}\right))$ , leading to non-Fickian diffusion. In section 4 we will also show that, for water vapour  $(\alpha = w)$ , the Fickian diffusion result of Eq. 15 is a special case of the expression for  $F_w$  in the four-component system, in the limiting case that the water vapour flux density is much greater than that for oxygen or carbon dioxide (i.e.,  $F_w \gg F_o$  or  $F_c$ ), which is a realistic assumption. For oxygen ( $\alpha = 0$ ), however, this is not the case (section 4).

As anticipated from Eq. 11, when  $x_{\alpha}$  is much smaller than 1 (i.e., for a dilute tracer), the contribution of Stefan flow (Eq. 16) becomes negligible, and  $F_{\alpha}$  is dominated by the diffusive component  $f_{\alpha}$  (Eq. 17). Therefore, the dependence of the effective diffusion coefficient on  $x_{\alpha}$  can be interpreted as a correction to Eq. 17 arising from the influence of Stefan flow. This correction is significant for O<sub>2</sub>. For CO<sub>2</sub>, the correction is smaller but still non-negligible, as demonstrated in the next section, where we apply the full derivation of the Stefan-Maxwell equations (Eq. 14) to the four-component system under study.

#### Ouaternary mixture: water vapour, oxygen and carbon dioxide diffusing in stagnant nitrogen

We now apply the Stefan–Maxwell equations (Eq. 14) to a four-component mixture consisting of water vapour, oxygen, and carbon dioxide diffusing in stagnant nitrogen (labelled by species indices  $\alpha = w, o, c$ , and n, respectively). Figure 1 shows the flux directions of the four components (Figure 1a), their orders of magnitude, and the decomposition of the O<sub>2</sub>-flux into a Stefan flow and a diffusive molar flux component (Figure 1b). If we consider the flux components in Figure 1b from the perspective of the physiological constraints,  $f_o$  depends on the photosynthesis rate and stomatal conductance.  $s_o$  depends on the transpiration rate, that is on water vapour pressure deficit and on stomatal conductance. If the conductance decreases, both  $f_o$  and  $s_o$  decrease, too.

In Eq. 14, for each species  $\alpha$  there are now three terms in the sum over all other species  $\beta \neq \alpha$  on the right-hand side. For  $H_2O$ ,  $O_2$  and  $CO_2$  ( $\alpha = w, o, c$ ) we obtain the following expressions for their mole fraction gradients in terms of the stomatal-scale molar flux densities:

$$\frac{\partial x_w}{\partial l} = \frac{x_w F_o - x_o F_w}{c_t D_{ow}} + \frac{x_w F_n - x_n F_w}{c_t D_{wn}} + \frac{x_w F_c - x_c F_w}{c_t D_{wc}}$$

$$(18a)$$

$$\frac{\partial x_o}{\partial l} = \frac{x_o F_w - x_w F_o}{c_t D_{ow}} + \frac{x_o F_n - x_n F_o}{c_t D_{oo}} + \frac{x_o F_c - x_c F_o}{c_t D_{oc}}$$

$$\frac{\partial x_c}{\partial l} = \frac{x_c F_o - x_o F_c}{c_t D_{oc}} + \frac{x_c F_n - x_n F_c}{c_t D_{cn}} + \frac{x_c F_w - x_w F_c}{c_t D_{wc}}.$$
(18b)

$$\frac{\partial x_c}{\partial l} = \frac{x_c F_o - x_o F_c}{c_t D_{oc}} + \frac{x_c F_n - x_n F_c}{c_t D_{cn}} + \frac{x_c F_w - x_w F_c}{c_t D_{wc}}.$$
(18c)

#### 4.1 Non-Fickian relationship between H<sub>2</sub>O and O<sub>2</sub> gradients and fluxes 200

The derived equations relating gradients to multiple fluxes (Eqs. 18a-18c) simplify if we assume that (1) on stoichiometric grounds the  $CO_2$  molar flux density is equal and opposite to the  $O_2$  molar flux density ( $F_c = -F_o$ ), and (2)  $N_2$  is stagnant  $(F_n=0)$ . Focusing now on  $H_2O$  and  $O_2$ , and making the substitutions  $F_c=-F_o$ ,  $F_n=0$  and  $x_n=1-x_o-x_w-x_c$ , Eqs. 18a-18c can be written in the form:

Figure 1. Multicomponent fluxes in a system composed of water vapour (w), oxygen (o), and carbon dioxide (c) diffusing in stagnant nitrogen (n). The mole fraction at the sub-stomatal cavity and the atmosphere are expressed by the subscripts i and a, respectively. (a) Direction of the fluxes of the four compounds as investigated in our study, with the dominant role by the water vapor molar flux. (b) Decomposition of O2-flux into a Stefan flow component and a diffusive molar flux component. The dashed lines indicate that the Stefan flow and diffusive flux are much larger than the  $O_2$  flux as shown by the theory. L is the length and r the radius of the cylindrical symmetric stomatal pore, respectively. The O2 flux is prescribed based on observations of the CO2 flux at leaf level. More details on the geometry can be found at Fig. В1

205 
$$\frac{\partial x_w}{\partial l} = \frac{1}{c_t} (AF_w + BF_o)$$

$$\frac{\partial x_o}{\partial l} = \frac{1}{c_t} (CF_w + DF_o),$$
(19a)

$$\frac{\partial x_o}{\partial l} = \frac{1}{c_*} (CF_w + DF_o),\tag{19b}$$

where

$$A = -\frac{x_o}{D_{ow}} - \frac{1 - x_o - x_w - x_c}{D_{wn}} - \frac{x_c}{D_{wc}}$$
(20a)

$$B = x_w \left( \frac{1}{D_{ow}} - \frac{1}{D_{wc}} \right) \tag{20b}$$

$$210 \quad C = \frac{x_o}{D_{ow}} \tag{20c}$$

$$D = -\frac{x_w}{D_{ow}} - \frac{1 - x_o - x_w - x_c}{D_{on}} - \frac{x_o + x_c}{D_{oc}}.$$
 (20d)

The right-hand sides of Eqs. 19a-19b only depend on  $F_w$  and  $F_o$ . Eqs. 19a-19b can be inverted (e.g. using the standard rules of matrix algebra) to express the fluxes in terms of the gradients:

$$F_w = \frac{c_t}{AD - BC} \left\{ D \frac{\Delta x_w}{\Delta l} - B \frac{\Delta x_o}{\Delta l} \right\}$$
 (21a)

215 
$$F_o = \frac{c_t}{AD - BC} \left\{ -C \frac{\Delta x_w}{\Delta l} + A \frac{\Delta x_o}{\Delta l} \right\}.$$
 (21b)

Therefore, in general, each of the  $H_2O$  and  $O_2$  fluxes  $F_w$  and  $F_o$  depends on both gradients - and vice-versa, as seen from Eqs. 19a-19b - giving a non-Fickian diffusion law between fluxes and gradients.

This is a general characteristics of the Stefan-Maxwell equations (Eq. 14), whether expressed in terms of  $F_{\alpha}$  or  $f_{\alpha}$ . The physical interpretation is that while  $\partial x_{\alpha}/\partial l$  represents the thermodynamic force driving the diffusion of species  $\alpha$  in the absence of other species, within a mixture this force is counterbalanced by the total drag force exerted on species  $\alpha$  by all other species. This drag force depends on the relative velocities —and consequently on the flux densities— of all species in the mixture. The Stefan-Maxwell equations, whether expressed in terms of  $F_{\alpha}$  or  $f_{\alpha}$ , describe this balance of forces. As a result, when solving for  $F_{\alpha}$  or  $f_{\alpha}$  by inverting the Stefan-Maxwell equations,  $F_{\alpha}$  or  $f_{\alpha}$  depend on the mole fraction gradients of all species, not just on  $\partial x_{\alpha}/\partial l$ .

220

However, in the present case, Eqs. 21a-21b simplify further if we note that typically  $F_w$  is much larger than  $F_o$ , so that in the right-hand sides of Eqs. 19a-19b the terms involving  $F_o$  are much smaller than the terms involving  $F_w$ , and to a first approximation can therefore be neglected, leading to

$$\frac{\partial x_w}{\partial l} \approx \frac{1}{c_t} A F_w \approx \frac{1 - x_w}{c_t D_{wa}} F_w \tag{22a}$$

$$\frac{\partial x_o}{\partial l} \approx \frac{1}{c_t} C F_w \approx \frac{x_o}{c_t D_{wa}} F_w,$$
 (22b)

where we have substituted the terms A and C using Eqs. 20a and 20b. For the second equalities in Eqs. 22a-22b we have made the further approximations in Eq. 20a: (1)  $x_c$  is typically much smaller than  $x_o$  and  $x_w$ , and (2)  $D_{wn} \approx D_{ow} \approx D_{wa}$  (Table A1).

Eq 22a reproduces the same Fickian relationship between the  $H_2O$  flux and  $H_2O$  gradient that was obtained previously for the binary case of  $H_2O$  diffusing in stagnant air (Eq. 15 with  $\alpha = w$ ). In contrast, Eqs. 22b implies that the  $O_2$  gradient is proportional to the  $H_2O$  flux, and not the  $O_2$  flux.

#### 4.2 Diffusive and molar flows

As shown by Figure 1b is interesting to calculate the contribution of the Stefan flow and diffusive molar flux to the flux. Using Eqs. 7, 11 and 12 we can quantify the Stefan flow and diffusive components of the molar flux densities for a given species  $\alpha$ . In the quaternary system, since  $F_c + F_o = F_n = 0$  - recall that we assume  $F_c = -F_o$  -, the total molar flux density of the mixture ( $F_t$ ) (Eq. 7) is therefore equal to the water flux density, *i.e.*.  $F_t = F_w$ . From Eqs. 11 and 12, therefore:

$$s_{\alpha} = x_{\alpha} F_{w} \tag{23}$$

and

$$f_{\alpha} = F_{\alpha} - x_{\alpha} F_{w} \tag{24}$$

These relationships between the stomatal-scale flux densities  $s_{\alpha}$ ,  $f_{\alpha}$  and  $F_{\alpha}$  (mol m<sup>-2</sup> pore s<sup>-1</sup>) also apply to the leaf-scale flux densities. Here we use the following conversions as follows simply by multiplying both sides of Eqs. 23-24 by  $a_s \rho_s$ :  $s_{\alpha}^{leaf} = a_s \rho_s s_{\alpha}$ ,  $f_{\alpha}^{leaf} = a_s \rho_s f_{\alpha}$  and  $F_{\alpha}^{leaf} = a_s \rho_s F_{\alpha}$  (mol m<sup>-2</sup> leaf s<sup>-1</sup>), where  $a_s$  (m<sup>2</sup> pore) is the cross-sectional area of the stomatal aperture and  $\rho_s$  (number of stomata m<sup>-2</sup> leaf) is stomatal leaf density.

This allows us to gauge the relative importance of the Stefan and diffusive components at leaf scales using representative values of  $F_{\alpha}^{leaf}$ . Table 1 gives values of  $s_{\alpha}^{leaf}$  and  $f_{\alpha}^{leaf}$  for H<sub>2</sub>O, O<sub>2</sub> and CO<sub>2</sub>. Also shown are the corresponding scaled gradients  $a_s\rho_s\partial x_{\alpha}/\partial l$  calculated by multiplying both sides of Eqs. 18a-18c by  $a_s\rho_s$  and substituting  $F_{\alpha}^{leaf}$  into the right-hand sides of Eqs. 18a-18c.

Table 1.

Stefan flow  $(s_{\alpha}^{leaf})$  and diffusive molar flux density  $(f_{\alpha}^{leaf})$  components of the leaf-scale  $H_2O$ ,  $O_2$  and  $CO_2$  molar flux densities  $(F_{\alpha}^{leaf})$ , calculated by multiplying both sides of Eqs. 23 and 24 by  $a_s\rho_s$  and substituting typical values of  $F_{\alpha}^{leaf}=a_s\rho_sF_{\alpha}$ , where  $a_s$  (m<sup>2</sup> pore) is stomatal aperture,  $\rho_s$  (number of stomata m<sup>-2</sup> leaf) is stomatal leaf density and  $F_{\alpha}$  is the stomatal-scale flux density. Also shown are the scaled mole fraction gradients  $a_s\rho_s\partial x_{\alpha}/\partial l$  calculated from Eqs. 18a-18c (multiplied through by  $a_s\rho_s$ ) using typical values of  $F_{\alpha}^{leaf}$  and the parameter values in Table A1. We use the following mole fractions for  $H_2O=1\cdot 10^{-2}$ ,  $O_2=0.21$  and  $CO_2=4\cdot 10^{-4}$ . For this value of  $H_2O$ , it corresponds to a specific humidity equal to 16.1 gr<sub>w</sub> kg<sub>air</sub><sup>-1</sup> (representative high atmospheric humid conditions like the tropical forest).

|                      | Molar flux density,                         | Stefan flow,                                       | Diffusive flux density                             | Scaler gradient                            |
|----------------------|---------------------------------------------|----------------------------------------------------|----------------------------------------------------|--------------------------------------------|
|                      | $F_{\alpha}^{leaf} = a_s \rho_s F_{\alpha}$ | $s_{\alpha}^{leaf} = a_s \rho_s s_{\alpha}$        | $f_{\alpha}^{leaf} = a_s \rho_s f_{\alpha}$        | $a_s \rho_s(\partial x_\alpha/\partial l)$ |
|                      | [                                           | [ $\mu$ mol m <sup>-2</sup> leaf s <sup>-1</sup> ] | [ $\mu$ mol m <sup>-2</sup> leaf s <sup>-1</sup> ] | [m <sup>-1</sup> ]                         |
| H <sub>2</sub> O (w) | $10^{4}$                                    | 100                                                | 9900                                               | -9.37                                      |
| O <sub>2</sub> (o)   | 10                                          | 2100                                               | -2090                                              | 1.94                                       |
| CO <sub>2</sub> (c)  | -10                                         | 4                                                  | -14                                                | 0.016                                      |

As shown in Eq. 22a the gradient of  $H_2O$  is also dependent of the mole fraction of water. Under very dry conditions, for instance air masses characterized by mole fractions  $H_2O = 10^{-3}$  equal to the specific humidity of 1.6  $g_w$   $kg_{air}^{-1}$ , the scalar gradient is -9.46 m<sup>-1</sup> compared to the control value of -9.37 <sup>-1</sup> m<sup>-1</sup> as shown in Table 1. These results stress the already-noted importance of Stefan flow for  $O_2$  transport relative to Stefan flow for  $H_2O$  and  $CO_2$ . For  $H_2O$ , characterized by a positive value (*i.e.* stomata to atmosphere) Stefan flow (100 mol m<sup>-2</sup> leaf s<sup>-1</sup>) is dominated by an even larger diffusive flow in the same direction (9900 mol m<sup>-2</sup> leaf s<sup>-1</sup>). For  $CO_2$ , the positive Stefan flow (4 mol m<sup>-2</sup> leaf s<sup>-1</sup>) is offset by diffusion in the opposite direction (-14 mol m<sup>-2</sup> leaf s<sup>-1</sup>), but yet relevant. For  $O_2$  the diffusive and Stefan components are individually two orders of magnitude larger than the prescribed  $F_0$  but are almost equal in magnitude and opposite in direction to each other.

Another consequence of the relative importance of Stefan flow in  $O_2$  transport is that, unlike  $H_2O$  and  $CO_2$ , the flux of  $O_2$  does not necessarily follow the opposite direction of its mole fraction gradient. Specifically, both  $F_w$  and  $F_c$  are opposite in sign to their respective mole fraction gradients, *i.e.* following qualitatively the Fickian law relationship between the flux and the gradient. This means that water vapour and  $CO_2$  are transported "downhill", from regions of high to low mole fractions, although deviating from Fick's law due to the influence of the fluxes of other species.

This is not the case for  $O_2$ . As shown in Eq. 22b, the  $O_2$  gradient is directly proportional to, and in the same direction as, the  $O_2$  Stefan flow rate,  $s_o = x_o F_w$ . This represents a clear departure from classical Fickian diffusion, where flux and gradient are expected to have opposite direction. We explore the conditions for uphill  $O_2$  transport more generally in Section 4.4.

#### 4.3 Relationship between internal O<sub>2</sub> mole fraction and relative humidity

From the gradient-flux relationships given by Eqs. 22a-22b (which are based on the realistic approximations  $F_w \gg F_o$ ,  $x_c \ll x_o$  and  $x_w$ , and  $D_{wn} \approx D_{ow} \approx D_{wa}$ ), it follows that the gradients in  $O_2$  and  $O_2$  and  $O_2$  satisfy the relationship

$$\frac{1}{x_o}\frac{\partial x_o}{\partial l} = -\frac{1}{1 - x_w}\frac{\partial x_w}{\partial l}.$$
 (25)

By integrating Eq. 25 from l = 0 (internal stomata) to l = L (atmosphere, see Fig. B1), it follows that

$$\frac{x_{oi}}{x_{oa}} \approx \frac{1 - x_{wi}}{1 - x_{wa}} \approx \frac{1 - \eta \, q_s}{1 - \eta \, q_s \, RH},\tag{26}$$

where  $x_{oi}$  and  $x_{oa}$  are, respectively, the internal and atmospheric  $O_2$  mole fractions,  $x_{wi}$  and  $x_{wa}$  being the corresponding H<sub>2</sub>O mole fractions. In the second equality we have expressed the latter in terms of the atmospheric saturated specific humidity  $(q_s)$ , the atmospheric relative humidity RH (assuming RH = 1 internally), and the air-to-water molecular weight ratio  $(\eta = 1.61kg_akg_w^{-1})$  to convert from mole fractions to relative humidity in expression 26.

From the expression 26 we infer that from normal atmospheric values of relative humidity (RH < 1), it follows from Eq. 26 that the internal  $O_2$  mole fraction is smaller than its atmospheric value. Figure 2 shows the predicted ratio of internal to atmospheric  $O_2$  mole fractions as a function of relative humidity. For all the values of RH, the mole fraction in the substomatal cavity is lower than the one of the atmosphere. The lowest internal  $O_2$  mole fraction occurs when RH = 0 (largest water demand by the atmosphere), when the predicted internal  $O_2$  fraction is approximately 3% lower than its atmospheric value. The predicted internal and atmospheric  $O_2$  mole fraction equalise at RH = 1. However, when RH = 1 (lowest water demand) the assumption  $F_w \gg F_o$  that underlies Eq. 26 is no longer valid. As a result, the simple relationship of Eq. 26 breaks down, and Eq. 22b must be replaced by the exact expression given by Eq. 19b, i.e., including the term  $DF_o$ ; since the coefficient D is negative (Eq. 20d). The effect of including this term is to decrease the  $O_2$  gradient (i.e., increase the ratio  $x_{oi}/x_{oa}$ ) so that the internal and atmospheric  $O_2$  mole fractions equalize at a value of RH slightly less than 1 (data not shown).

#### 4.4 Uphill vs downhill O<sub>2</sub> transport: three regimes

280

In the previous sections, we have discussed the dominant role of the  $H_2O$  flux with respect the  $O_2$  transport. Here, we examine more generally the conditions under which  $O_2$  transport ( $F_o$ ) and/or its diffusive component ( $f_o$ ) occurs 'uphill', i.e., against the  $O_2$  mole fraction gradient, due to the dominant drag-force effect of water-driven Stefan flow. For  $F_o > 0$  (i.e.,  $O_2$  transport in the direction stoma to atmosphere), uphill  $O_2$  transport occurs when  $\partial x_o/\partial l > 0$ . From the exact expression for the  $O_2$  gradient given by Eq. 19b, this occurs when  $CF_w + DF_o > 0$ . The expression reads:

Figure 2. The ratio of the internal oxygen to the atmospheric oxygen mole fraction as a function of the relative humidity calculated using Eq. 26, and a saturation specific humidity of  $q_s = 19.6 \cdot 10^{-3} \text{ kg}_w \text{ kg}_a^{-1}$  calculates at the temperature 298.15 K

$$\frac{F_w}{F_o} > \frac{-D}{C} = \frac{x_w}{x_o} + \frac{D_{ow}}{D_{on}} \frac{1 - x_o - x_w - x_c}{x_o} + \frac{D_{ow}}{D_{oc}} \frac{x_o + x_c}{x_o} = \frac{1}{x_o} (1 + R)$$
 (27a)

(27b)

$$(27d)$$

$$R = \left(\frac{D_{ow}}{D_{on}} - 1\right) (1 - x_o - x_w - x_c) + \left(\frac{D_{ow}}{D_{oc}} - 1\right) (x_o + x_c). \tag{27e}$$

Since  $D_{ow} > D_{on}$  and  $D_{ow} > D_{oc}$  (Table 1) it follows that R > 0. With the assumed values of  $F_w$ ,  $F_o$  and parameter values in Table 1, the condition Eq. 27a is comfortably satisfied, the left- and right-hand sides being 100 and  $\approx 5.5$  respectively. This condition will be violated when transpiration is sufficiently small compared to  $O_2$  transport (e.g. when RH is close to 1).

From Eq. 24 with  $\alpha=0$ , the diffusive  ${\rm O}_2$  flux density is negative  $(f_o<0)$  whenever

295

$$\frac{F_w}{F_o} > \frac{1}{x_o}. (28)$$

Thus, as the ratio of water to oxygen flux densities  $(F_w/F_o)$  increases, we can identify three oxygen transport regimes (Table 2), distinguished by the relative orientation of the  $O_2$  gradient  $(\partial x_o/\partial l)$ ,  $O_2$  flux density  $(F_o)$  and diffusive  $O_2$  flux density  $(f_o)$ .

Table 2.

Three  $O_2$  transport regimes distinguished by the relative orientations of the  $O_2$  gradient  $(\partial x_o/\partial l)$ ,  $O_2$  flux density  $(F_o)$ , assumed positive), and diffusive  $O_2$  flux density  $(f_o)$ , as determined by the value of the flux ratio  $(F_w/F_o)$  according to Eqs. 27a-27e and 28. Positive values of  $\partial x_o/\partial l$  indicate that the internal  $O_2$  mole fraction is less than atmospheric  $O_2$  mole fraction. Positive  $f_o$  or  $F_o$  show that flux direction is from stoma to the atmosphere.

| Regime                                                     | Sign of $\frac{\partial \mathbf{x_o}}{\partial \mathbf{l}}$ | Sign of f <sub>0</sub> ;        | Sign of F <sub>0</sub> ;        |
|------------------------------------------------------------|-------------------------------------------------------------|---------------------------------|---------------------------------|
|                                                            |                                                             | Direction relative to gradient  | Direction relative to gradient  |
| $1: \frac{F_w}{F_o} < \frac{1}{x_o}$                       | negative                                                    | positive: stomata -> atmosphere | positive: stomata -> atmosphere |
|                                                            |                                                             | Downhill                        | Downhill                        |
| 2: $\frac{1}{x_o} < \frac{F_w}{F_o} < \frac{1}{x_o} (1+R)$ | negative                                                    | negative: atmosphere -> stomata | positive: stomata -> atmosphere |
|                                                            |                                                             | Uphill                          | Downhill                        |
| $3: \frac{F_w}{F_o} > \frac{1}{x_o} (1+R)$                 | positive                                                    | negative: atmosphere -> stomata | positive: stomata -> atmosphere |
|                                                            |                                                             | Downhill                        | Uphill                          |

Table 2 outlines three regimes for water vapour and oxygen transport, defined by the direction of their respective mole fraction gradients. These regimes are characterised as uphill—where O<sub>2</sub> is transported from the stomata to the atmosphere despite an opposing mole fraction gradient—and downhill, where the flux and gradient are in opposite directions, consistent with classical Fickian diffusion. As noted above, due to the typically large value of the flux ratio  $(F_w/F_o)$ , and since  $(1+R)/x_o \approx 5.5$ , the  $O_2$  transport between the stoma and the atmosphere typically falls within regime 3, resulting in uphill transport of  $O_2$  ( $F_o$ ) in the direction from low to high O<sub>2</sub> mole fraction, as already seen in Table 2. In this regime, the molar flux density of H<sub>2</sub>O drags  $O_2$  molecules from the stoma to the atmosphere, driven by the  $O_2$  Stefan flow  $s_o = x_o F_w$ . This drag effect leads to lower  $O_2$  mole fraction in the stoma than in the atmosphere ( $x_{oi} < x_{oa}$ ), and a negative diffusive  $O_2$  molar flux density ( $f_o$ ) (i.e. from the atmosphere to the stomata) that counterbalances but does not completely offset positive Stefan flow (Table 2) since we prescribe that the  $O_2$  flux is from the stoma into the atmosphere. As  $F_w/F_o$  decreases into regime 2, the  $O_2$  gradient changes sign but the diffusive flux  $f_o$  remains negative (i.e.. Stefan flow is still important), resulting in uphill diffusion from low to high O<sub>2</sub> mole fractions; this reflects the non-Fickian character of the flux-gradient relationships in general. At very low values of  $F_w/F_o$  (regime 1), Stefan flow becomes very small,  $f_o$  changes sign, and  $O_2$  transport is dominated by positive diffusion from the stoma to the atmosphere. It is important to note that the characterization of these three regimes does not account for the roles of photorespiration or dark respiration, and potential sources of O<sub>2</sub> within the sub-stomatal cavity are omitted from the analysis. Similarly, transport processes driven by thermal differences between the stomata and the atmosphere (thermodiffusion) are omitted.

#### 5 Discussion

Our research extends the studies of Jarman (1974) and von Caemmerer and Farquhar (1981), which examined the transport of a ternary gas mixture comprising air, water vapour, and carbon dioxide. Here, air is composed by combining O<sub>2</sub> and N<sub>2</sub>. Jarman (1974) also briefly considered the CO<sub>2</sub>-H<sub>2</sub>O-O<sub>2</sub>-N<sub>2</sub> quaternary system considered in our study, but only with the aim of verifying the insignificant effect of O<sub>2</sub> transport on internal CO<sub>2</sub> mole fraction. For completeness, in Appendix D we demonstrate that our equations for the quaternary system (Eqs. 18a-18c) reduce to these ternary formulations.

Following Jarman (1974), in Appendix E we evaluated the percentage corrections to the O<sub>2</sub> gradient due to Stefan molar flow, using a ternary system (CO<sub>2</sub>, H<sub>2</sub> and air, the latter compost by O<sub>2</sub> and N<sub>2</sub>) as the reference system (cf. section 3). As shown in Table E1, our results are consistent with those in Table 1 and confirm the key role of the H<sub>2</sub>O molar flux density in determining the O<sub>2</sub> gradient.

Although previous studies have recognized that transport of O<sub>2</sub> is distinguished by its higher mole fraction and that Stefan flow could be relevant across scales from the canopy to the atmospheric boundary layer (Kowalski et al., 2021), our analysis of the fundamental molar balance equation demonstrates that the primary effects of Stefan flow emerge at the stomatal scale. To this end Appendix C shows a non-dimensional analysis of the governing equation of O<sub>2</sub> calculating the O<sub>2</sub>-transport under representative Reynolds numbers measured in a greenhouse and at the top of a rainforest canopy. Connecting to previous studies of O<sub>2</sub> transport, Kowalski (2024) has proposed a simple model for the effects of water vapour flow on O<sub>2</sub> transport which assumes, a priori and without rigorous physical foundation, that the sub-stomatal O<sub>2</sub> pressure deficit (relative to the atmosphere) is equal to the atmospheric O<sub>2</sub> mole fraction multiplied by the water vapour pressure surplus in the sub-stomatal cavity. Since partial pressure is proportional to mole fraction, this assumption implies the relationship.

$$x_{oa} - x_{oi} = x_{oa}(x_{wi} - x_{wa}), (29)$$

Interestingly, Eq. 26 of our study is equivalent to the relationship

345 
$$x_{oa} - x_{oi} = \frac{x_{oa}}{1 - x_{wa}} (x_{wi} - x_{wa}),$$
 (30)

which, because  $x_{wa} \ll 1$ , is practically identical to Eq. 29. However, in contrast to Eq. 29, our result (Eq. 30) has been derived from the fundamental physics of multicomponent diffusion (Stefan-Maxwell equations), and is itself an approximation (which assumes in particular that  $F_w \gg F_o$ ). Our underlying theory not only provides a physical justification for Eq. 29, but also a rigorous framework for calculating corrections to it under a wider range of conditions.

Our study does not yet address the physiological connection between sources and sinks of O<sub>2</sub> and CO<sub>2</sub> driven by photosynthesis, photorespiration and dark respiration. A logical next step would be to couple our theoretical expressions for the non-Fickian relationship between gradients and fluxes to biochemical models of photosynthetic CO<sub>2</sub> fixation (e.g. (Farquhar, 1982)) and O<sub>2</sub> transport dynamics during photosynthesis and photorespiration. Traditional models often over-simplify pho-

torespiration and dark respiration by neglecting the dynamic interactions between different gases, key processes in the transport stomata-atmosphere.

Our theoretical findings — particularly the influence of  $H_2O$  molar flux density on internal  $O_2$  mole fraction — suggest that incorporating a more physically realistic representation of  $O_2$  dynamics could improve the coupling with models of photosynthetic processes, leading to more accurate estimates of  $O_2$  transport's impact on photosynthetic efficiency. A better understanding of these processes could refine the the predictions of photosynthesis under varying environmental conditions (Zhang et al., 2024), where photorespiration critically limits carbon assimilation. In a similar vein, it will be convenient to study the relationship between the  $O_2$  flux density with respect to the  $CO_2$  flux relationship. In our study, we assume on stoichiometric grounds that the  $O_2$  flux density is equal and opposite to the  $CO_2$  molar flux density. However, we do not have observational evidence that this occurs at the leaf level. Besides the Stefan (molar) flow, additional processes such as thermodiffusion (Curtiss and Hirschfelder, 1949; Griffani et al., 2024) may also contribute to gas transport, particularly in the presence of leaf—air temperature gradients. In the present work, we did not account for these thermal effects, but they may become relevant for leaves exposed to strong radiation loads or under fluctuating microclimatic conditions, such as cloud passages or sunflecks (van Diepen et al., 2025). An open question remains regarding the relative importance of thermodiffusion in comparison to Stefan flow and molecular diffusion. To this end, a future aim is to represent  $O_2$  biophysical processes independently from  $CO_2$  at all relevant atmospheric scales.

One key prediction in our study — that the internal stomatal O<sub>2</sub> mole fraction is typically lower than the atmospheric O<sub>2</sub> mole fraction (Eq. 26, Fig. 2) — offers an opportunity to experimentally test the underlying multicomponent diffusion theory on which it is based. Such a test would require the precise measurement of internal O<sub>2</sub> mole fraction under stagnant or low Reynolds number atmospheric conditions and without photosynthesis. Fig. 2 suggests that such measurements would be best carried out under a wide range of relative humidity, and expecting the larger gradient values under low relative humidity values. Experimental deviations from the predictions shown in Fig. 2 would also be of value, because they would signal missing physics not included in our current approach - which assumes the ideal gas approximation and isothermal conditions -; crucially, the underlying theory of the Stefan-Maxwell equations provides a rigorous thermodynamic framework for including such extensions (e.g. to non-ideal gases and thermal forces).

#### 6 Conclusions

Oxygen transport between the sub-stomatal cavity and the atmosphere exhibits two key characteristics that require particular consideration: (i) the relatively high mole fraction of oxygen compared to other primary components of the photosynthesis system, and (ii) the significant influence of the water molar flux density (transpiration) on oxygen transport. The Stefan-Maxwell equations provided a rigorous physical framework for examining those characteristics quantitatively. Based on these equations, our findings emphasize the interplay between fluxes, gradients, and diffusivity coefficients within the quaternary N<sub>2</sub>-O<sub>2</sub>-H<sub>2</sub>O-CO<sub>2</sub> system, and provide new insights into the dynamics of oxygen stomatal gas exchange. The main conclusions are:

- 1. The water vapour molar flux density generates a drag force that drives the movement of oxygen from the stomatal interior to the atmosphere. The large water-driven Stefan or molar flow of oxygen is almost (but not exactly) counterbalanced by a diffusive O<sub>2</sub> flux from the atmosphere to the stomatal interior.
- 2. We identify three distinct regimes of O<sub>2</sub> transport depending on the ratio of water to oxygen molar flux densities. These regimes are distinguished by the relative orientations of the O<sub>2</sub> molar flux density, the diffusive component, and gradient in O<sub>2</sub> mole fraction between the sub-stomatal cavity and the atmosphere. Under typical conditions, oxygen moves uphill from stoma to the atmosphere from low to high O<sub>2</sub> mole fraction, whereas the diffusive component of O<sub>2</sub> transport moves from the atmosphere to the stomata (downhill).
- 39. For this typical regime, we derived a simple analytical expression for the ratio of the internal to atmospheric O<sub>2</sub> mole fractions as a function of relative humidity, which explicitly accounts for the drag effect of water-driven Stefan flow. This expression offers the possibility to interpret and explain experiments measuring simultaneous O<sub>2</sub> mole fractions in the stomata and the atmosphere, grounded in underlying multicomponent diffusion theory. If future experiments indeed, measure a lower stomatal internal oxygen mole fraction compared to the atmosphere, our results would offer a robust physical explanation.

Our theoretical results establish a baseline for developing a general multicomponent theory of stomatal gas exchange, which can also be extended to other carbon tracers, such as stable isotopologues. By incorporating isotopic variants of CO<sub>2</sub> (e.g.,  $^{13}$ CO<sub>2</sub>,  $^{C18}$ OO), the framework can help disentangle the relative contributions of different biochemical and physical processes—such as diffusion, carboxylation, and photorespiration—to overall gas exchange. This extension opens new avenues for interpreting isotopic measurements in both laboratory and field settings, providing a physically sounded representation of the transport of carbon tracers across scales and more comprehensive understanding of leaf-level carbon and oxygen dynamics under varying environmental conditions.

405

### Appendix A: List of variables, symbols, values and units

Table A1 presents the definitions of the variables used in this study, including their symbols, units, and characteristic values applied in the calculations. The calculations are performed at the stomatal level, with units of [mol s<sup>-1</sup>], while the input variables are based on representative observations taken at the leaf level, where the units are [mol m<sup>-2</sup> s<sup>-1</sup>] (Jarman, 1974; Vilà-Guerau de Arellano et al., 2020). Leaf-level values are downscaled to the stomatal level by dividing them by the leaf stomatal density, which represents the number of stomata per unit area of leaf. "water" is referring to water vapour.

Table A1: Definitions of the variables, symbols, characteristic values, and units used in the study. Molecular diffusivities are calculated at 293 K using the expression proposed by Reid et al. (1987)

| Variable                                                                          | Symbol                      | Value                 | Units                                                    |
|-----------------------------------------------------------------------------------|-----------------------------|-----------------------|----------------------------------------------------------|
| Molecular weight of dry air                                                       | M                           | $2.89 \cdot 10^{-3}$  | kg mol <sup>−1</sup>                                     |
| Density of dry air                                                                | $ ho_a$                     | $1.204 \cdot 10^{-3}$ | ${\rm kg}~{\rm m}^{-3}$                                  |
| Mole density of species mixture                                                   | $c_t = \rho_a / M$          | 41.58                 | $\mathrm{mol}\;\mathrm{m}^{-3}$                          |
| Mole fraction of species $\alpha$                                                 | $c_{\alpha}$                | -                     | $\mathrm{mol}\;\mathrm{m}^{-3}$                          |
| Mole fraction of species $\alpha$                                                 | $x_{\alpha}$                | -                     | mol mol <sup>−1</sup>                                    |
| Mole fraction atmospheric of CO <sub>2</sub>                                      | $x_c$                       | $4.10^{-4}$           | $\mathrm{mol}\ \mathrm{mol}^{-1}$                        |
| Mole fraction of atmospheric O <sub>2</sub>                                       | $x_{o_a}$                   | $21 \cdot 10^{-2}$    | $\mathrm{mol}\ \mathrm{mol}^{-1}$                        |
| Mole fraction of internal O <sub>2</sub>                                          | $x_{o_i}$                   | calculated            | $\mathrm{mol}\ \mathrm{mol}^{-1}$                        |
| Mole fraction water vapour                                                        | $x_w$                       | $1.10^{-2}$           | $\mathrm{mol}\ \mathrm{mol}^{-1}$                        |
| Internal specific humidity                                                        | $q_i$                       | -                     | $kg_w kg_a^{-1}$                                         |
| Atmospheric humidity atmosphere                                                   | $q_a$                       | -                     | $kg_w kg_a^{-1}$                                         |
| Saturated specific humidity                                                       | $q_{sat}$                   | -                     | $kg_w kg_a^{-1}$                                         |
| Relative humidity of the atmosphere                                               | rh                          | calculated            | -                                                        |
| molar flux density of species $\alpha$ (leaf)                                     | $F_{lpha}$                  | -                     | $\mu \mathrm{mol} \; \mathrm{m}^{-2} \; \mathrm{s}^{-1}$ |
| Diffusive molar flux density                                                      | $f_{lpha}$                  | calculated            | $\mu \mathrm{mol} \; \mathrm{m}^{-2} \; \mathrm{s}^{-1}$ |
| Stefan flow (convective like molar flux density) of species $\boldsymbol{\alpha}$ | $s_{\alpha} = c_{\alpha} u$ | calculated            | $\mu \mathrm{mol} \; \mathrm{m}^{-2} \; \mathrm{s}^{-1}$ |
| Molar flux density of water                                                       | $F_w$                       | observation           | $\mu \mathrm{mol} \; \mathrm{m}^{-2} \; \mathrm{s}^{-1}$ |
| Molar flux density of CO <sub>2</sub>                                             | $F_c$                       | observation           | $\mu \mathrm{mol} \; \mathrm{m}^{-2} \; \mathrm{s}^{-1}$ |
| Molar flux density of O <sub>2</sub>                                              | $F_o$                       | inferred $F_c$        | $\mu \mathrm{mol} \; \mathrm{m}^{-2} \; \mathrm{s}^{-1}$ |
| Molar flux density of N <sub>2</sub>                                              | $F_N$                       | 0                     | $\mu \mathrm{mol} \; \mathrm{m}^{-2} \; \mathrm{s}^{-1}$ |
| Binary diffusivity of species $\alpha$ and $\beta$                                | $\mathrm{D}_{lphaeta}$      | -                     | $m^2 s^{-1}$                                             |
| Molecular diffusivity water to oxygen <sup>1</sup>                                | $\mathbf{D}_{ow}$           | $2.58 \cdot 10^{-5}$  | $m^2 s^{-1}$                                             |
| Molecular diffusivity nitrogen to water <sup>1</sup>                              | $D_{wn}$                    | $2.53 \cdot 10^{-5}$  | $m^2 s^{-1}$                                             |
|                                                                                   |                             | Cor                   | ntinued on next page                                     |

Table A1 – continued from previous page

| Variable                                              | Symbol                | Value                | Units                    |
|-------------------------------------------------------|-----------------------|----------------------|--------------------------|
| Molecular diffusivity nitrogen to oxygen <sup>1</sup> | $D_{on}$              | $2.02 \cdot 10^{-5}$ | ${ m m}^2 { m s}^{-1}$   |
| Molecular diffusivity water to carbon <sup>1</sup>    | $\mathrm{D}_{wc}$     | $2.05 \cdot 10^{-5}$ | ${\rm m}^2~{\rm s}^{-1}$ |
| Molecular diffusivity oxygen to carbon <sup>1</sup>   | $\mathrm{D}_{oc}$     | $1.59 \cdot 10^{-5}$ | ${\rm m}^2~{\rm s}^{-1}$ |
| Molecular diffusivity carbon to nitrogen <sup>1</sup> | $\mathrm{D}_{cn}$     | $1.59 \cdot 10^{-5}$ | ${ m m}^2 { m s}^{-1}$   |
| Mole-averaged velocity                                | u                     | -                    | ${ m m~s^{-1}}$          |
| Mole velocity of tracer $\alpha$                      | $\mathfrak{u}_{lpha}$ | -                    | ${ m m~s^{-1}}$          |
| One-dimensional direction length                      | 1                     | -                    | m                        |
| Representative length scale stomata-atmosphere        | L                     | calculated           | m                        |

<sup>&</sup>lt;sup>1</sup> All the molecular diffusivities are calculated Reid et al. (1987)

415

### Appendix B: Geometry and scaling of one-dimensional gas exchange between stomata and the atmosphere

Fig. B1 shows the one-dimensional representation of the stomata and the atmosphere in which the Stefan-Maxwell equations are applied. The transport of all the species is through the stomatal pore that connects between the sub-stomatal cavity and the atmosphere. This geometry and assumptions are similar to the binary and quaternary mixtures (see Figure 1).

#### 420 Appendix C: Non-dimensional O<sub>2</sub> governing equation including Stefan flow effects

A key question in our analysis is to determine when a multicomponent diffusion representation is necessary to accurately describe oxygen transport. To address this, we consider the one-dimensional molar balance equation for  $O_2$  and non-dimensionalize it using characteristic length and velocity scales. For the binary system of  $O_2$  diffusing in stagnant  $N_2$  (section 3), Eqs. 13 becomes:

425 
$$\frac{\partial x_o}{\partial t} = -\frac{\partial (x_o u)}{\partial l} + \frac{\partial}{\partial l} \left( D_{no} \left( \frac{dx_o}{dl} \right) \right) + r_o.$$
 (C1)

in which we use Eq. 17 to represent the diffusive molar flux density  $(f_o)$  as a function of the gradient of oxygen  $(\frac{dx_o}{dl})$ . This equation has units of  $(\text{mol}_o \text{ mol}_{air}^{-1}) s^{-1}$ . Note that in the multicomponent diffusion approach, this equation must be solved simultaneously with the governing equations for u (Navier-Stokes equation) and for the mole fraction of water vapour  $x_w$ .

If Eq. C1 is made non-dimensional using the characteristic length  $\mathcal{L}$  and velocity  $\mathcal{U}$  inertial scales, the molecular diffusion 430 term will be divided by the Reynolds number, defined as the ratio of inertial acceleration to the molecular diffusion ( $\nu_a$ ) and it reads  $Re = \mathcal{L}\mathcal{U}/\nu_a$ ; and the Schmidt number, defined as the ratio of momentum diffusivity (kinematic viscosity) to mass

Figure B1. Sketch of the geometry of the stomata and atmosphere systems where the oxygen exchange takes place. The spatial one-dimensionality is represented by the coordinate l and the characteristic length scale L. Based on observations, we assume that this length scale is larger than the radius of the stomatal pore r, but it is not a theoretical requirement.

diffusivity ( $Sc = \nu_a/D_{no}$ ) (Vilà-Guerau de Arellano et al., 2004). To derive this non-dimensional equation, we introduce the following non-dimensional variables:  $\tilde{t} = (\mathcal{U}/\mathcal{L})t$ ,  $\tilde{u} = u/\mathcal{U}$ , and  $\tilde{l} = l/\mathcal{L}$ . Substituting these non-dimensional variables into Eq. C1 and dividing the equation by the factor ( $\mathcal{L}/\mathcal{U}$ ), we obtain the non-dimensional equation for oxygen. The final non-dimensional equation for oxygen is:

$$\frac{\partial x_o}{\partial \tilde{t}} = -\frac{\partial (x_o \tilde{u})}{\partial \tilde{l}} + \frac{\partial}{\partial \tilde{l}} \left( (ReSc)^{-1} \left( \frac{dx_o}{d\tilde{l}} \right) \right) + \tilde{r}_o, \tag{C2}$$

where the source term is  $\tilde{r}_o = r_o(\mathcal{L}/\mathcal{U})$ . The diffusive term (second term on the right-hand side) is now expressed as a function of the Reynolds number, defined as the ratio of inertial acceleration ( $Re = \mathcal{L}\mathcal{U}/\nu_a$ ), and the Schmidt number, defined as the ratio of momentum diffusivity (kinematic viscosity) to mass diffusivity ( $Sc = \nu_a/D_{no}$ ). Additionally, the source/sink term is made dimensionless by introducing a representative source/sink scale.

Using typical values for the flow velocity near the leaf surface (ranging from 0.05 to 0.15 m s<sup>-1</sup> taken by under greenhouse conditions Kimura et al. (2020), the length scale (order millimeters) and a kinematic viscosity of air  $1.5 \ 10^{-5} \ m^2 s^{-1}$ , we obtain Reynolds numbers on the order of 1 to 100. For larger leaves (0.1 m) and more realistic wind conditions — see, for instance, values of 1 m s<sup>-1</sup> taken at the canopy top in the Amazonian rain forest (González-Armas et al., 2025) — the Reynolds number is 6666. These greenhouse and forest Reynolds number values indicate that the flow remains laminar around the leaf surface.

As a result, we need to retain the molecular diffusion term (2nd right-hand side term) in the governing equation for  $O_2$ , and describe this  $O_2$  transport process using a multidiffusion component as described here.

#### **Appendix D: Equivalence ternary and quaternary systems**

In this Appendix, we study the equivalence between the equations derived for the quaternary system 18a, 18b and 18c, and the
450 ternary-system equations derived by Jarman (1974) and further applied by von Caemmerer and Farquhar (1981). We start by
rewriting the gradients of water and carbon dioxide as a function of the molar flux densities. The general expression for water
vapour is as follows:

$$\frac{\partial x_w}{\partial l} = \frac{1}{c_t} \left[ -\left(\frac{x_o}{D_{ow}} + \frac{x_n}{D_{wn}} + \frac{x_c}{D_{wc}}\right) F_w + \frac{x_w}{D_{ow}} F_o + \frac{x_w}{D_{wn}} F_n + \frac{x_w}{D_{wc}} F_c \right] \tag{D1}$$

and the gradient for carbon dioxide reads:

$$\frac{\partial x_c}{\partial l} = \frac{1}{c_t} \left[ -(\frac{x_o}{D_{ow}} + \frac{x_n}{D_{wn}} + \frac{x_w}{D_{wc}})F_c + \frac{x_c}{D_{oc}}F_o + \frac{x_c}{D_{cn}}F_n + \frac{x_c}{D_{wc}}F_w \right]$$
 (D2)

To connect and compare with expressions B9 and B10 at (von Caemmerer and Farquhar, 1981) we need to introduce the following assumptions: (1) in the ternary system used by Jarman (1974); von Caemmerer and Farquhar (1981), the fluxes  $F_o$  and  $F_n$  are grouped in a flux of air  $F_a$ . There is a mole fraction gradient of air across the stomata. However, they assume there net flux of air is zero, *i.e.*  $F_a$ = 0 at the interface by Jarman (1974); von Caemmerer and Farquhar (1981). (2) the mole fraction of oxygen and nitrogen are added  $x_o + x_n = x_a$ , (3) the diffusion coefficients are related as follow:  $D_{ow}$ = $D_{nw}$  and  $D_{oc}$ = $D_{nc}$ , and (4) we approximate the partial derivatives of  $x_o$ ,  $x_w$  and l with increments  $\Delta$ .

Using these previous assumptions, we define the conductance of water vapour in air, carbon dioxide in air, and carbon dioxide in water vapour as follows:

$$g_{aw} = \frac{c_t D_{wa}}{\Delta l} \quad g_{ac} = \frac{c_t D_{ca}}{\Delta l} \quad g_{ac} = \frac{c_t D_{wc}}{\Delta l}.$$
 (D3)

In applying these assumptions, the equations become the following:

$$\Delta x_w = -\left(\frac{x_a}{q_{aw}} + \frac{x_c}{q_{wc}}\right) F_w + \frac{x_w}{q_{wc}} F_c,\tag{D4}$$

and

$$\Delta x_c = -\left(\frac{x_a}{g_{ac}} + \frac{x_w}{g_{wc}}\right) F_c + \frac{x_c}{g_{wc}} F_w. \tag{D5}$$

These expressions are equivalent to the ones found by Jarman (1974) (see expressions of the CO<sub>2</sub>-gradient at page 932). However, to obtain the final equivalence with B9 and B10 at von Caemmerer and Farquhar (1981), we need to modify the notation. von Caemmerer and Farquhar (1981) defined water vapour and carbon dioxide gradients and fluxes in the following way:  $F_w = \Delta x_w = x_{wi} - x_{wa}$  and  $F_c \approx \Delta x_c = -x_{ca} + x_{ci}$ . Following this sign convention, we obtain:

$$x_{wi} - x_{wa} = \left(\frac{x_a}{g_{aw}} + \frac{x_c}{g_{wc}}\right) F_w + \frac{x_w}{g_{wc}} F_c,\tag{D6}$$

and

485

475 
$$x_{ci} - x_{ca} = -\left(\frac{x_a}{g_{ac}} + \frac{x_w}{g_{wc}}\right) F_c - \frac{x_c}{g_{wc}} F_w.$$
 (D7)

The late expressions are identical to B9 and B10 at (von Caemmerer and Farquhar, 1981) probing that the ternary solutions derived by Jarman (1974) can be retrieved from the more general fourth-equation systems here derived.

#### Appendix E: Quaternary versus binary mixture

From Eq. 18b, and after simplifying by assuming the net flux of nitrogen  $F_n = 0$ , we group the various terms as functions of the mole diffusive fluxes  $F_o$ ,  $F_w$  and  $F_o$ This allows us to derive the correction contributions to the single mole diffusive expression that follows Fick's law. Rearranging Eq. 18b, we obtain:

$$\frac{\partial x_o}{\partial l} = \frac{1}{c_t} \left[ -\left( \underbrace{\overrightarrow{x_n}}_{D_{on}} + \underbrace{\overrightarrow{x_w}}_{D_{ow}} + \underbrace{\overrightarrow{x_c}}_{D_{oc}} \right) F_o + \left( \underbrace{\overrightarrow{x_o}}_{D_{ow}} \right) F_w + \left( \underbrace{\overrightarrow{x_o}}_{D_o} \right) F_c \right] = \frac{1}{c_t} \left[ -(a+b+c)F_o + dF_w + eF_c \right]$$
 (E1)

Following the procedure outlined by Jarman (1974) to calculate the correction terms, we determine the percentage of the correction factor for the O<sub>2</sub>-gradient. We take term (a) as the reference, representing the interaction between oxygen molecules relative to the compound with the larger mole fraction—in this case, nitrogen. For the remaining terms, we assume that the correction factors are defined as follows:

$$C_1 = 100 \frac{b}{a}\% = 100 \frac{x_w D_{on}}{x_n D_{ow}}\%$$

$$C_2 = 100 \frac{c}{a}\% = 100 \frac{x_c D_{on}}{x_n D_{oc}}\%$$
(E2a)

$$C_{3} = -100\frac{d}{a}\% = -100\frac{x_{o}D_{on}}{x_{n}D_{ow}}\frac{F_{w}}{F_{o}}\%$$

$$C_{4} = -100\frac{e}{a}\% = -100\frac{x_{c}D_{on}}{x_{n}D_{oc}}\frac{F_{c}}{F_{o}}\%$$
(E2b)

Table E1 shows the correction values using the values of the molar flux densities and molecular diffusivity coefficients shown in Table A1. The very large value of the coefficient  $C_3$  corroborates the dominance of the Stefan flow.

As shown in Table E1 only  $C_3$  is the largest correction factor that strongly depends on the water vapour molar flux density. These results corroborate the key role played by the Stefan flow as as shown in Table 1.

#### Table E1.

Correction factors to the O<sub>2</sub> molar flux densities based on Eqs. E2a and E2b as a function of three different water vapour molar flux densities. The correction factors are dimensionless and in %. In bold the larger correction factors.

| $F_w \ [\mu \mathrm{mol} \ \mathrm{m}^{-2} \ \mathrm{leaf} \ \mathrm{s}^{-1}]$ | $C_1$              | $C_2$                                   | $C_3$             | $C_4$               |
|--------------------------------------------------------------------------------|--------------------|-----------------------------------------|-------------------|---------------------|
| $1.\cdot 10^4$                                                                 | $1.0 \cdot 10^{0}$ | $6.5 \cdot 10^{-2}$ $6.5 \cdot 10^{-2}$ | $-2.1 \cdot 10^4$ | $6.5 \cdot 10^{-2}$ |
| $1.\cdot 10^{3}$                                                               | $1.0 \cdot 10^{0}$ | $6.5 \cdot 10^{-2}$                     | $-2.1 \cdot 10^3$ | $8.1 \cdot 10^{-2}$ |
| $1.\cdot 10^{2}$                                                               | $1.0 \cdot 10^{0}$ | $6.5 \cdot 10^{-2}$                     | $-2.1 \cdot 10^2$ | $6.5 \cdot 10^{-2}$ |

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
