# Peer review of "New insights into stomatal oxygen transport viewed as a multicomponent diffusion process"

_EGUsphere, 2025_

## Referee Comment (RC1)

A review of "Technical Note: New insights into stomatal oxygen transport viewed as a multicomponent diffusion process" by Vila-Guerau de Arellano et al. (egusphere-2025-2705)

A. S. Kowalski

**Synopsis**

This study considers the drag exerted by water vapour molecules escaping stomata due to transpiration—the dominant gas exchange process—on oxygen molecules exported from stomata due to photosynthetic production. The issue at hand is the degree to which oxygen transport occurs down its concentration gradient due to Fickian diffusion, versus transport that may be a consequence of mass flow. The study is based on the broadly accepted theory of the Stefan-Maxwell equations, taken to represent the fundamental physics of multicomponent diffusion, and from which a multitude of equations are derived. The authors' mathematical manipulations appear to be correct; I hardly criticize them (but see Specific comments below). Rather, my doubts regarding these analyses concern their starting point, and specifically whether the Stefan-Maxwell equations do indeed accurately represent the fundamental physics of transport mechanisms.

**General assessment**

I neither endorse nor oppose publication of this manuscript. I have expressed my assessment that the concentration gradients determining diffusion should be specified, not as molar fractions (as in the Stefan-Maxwell equations) which seem to violate Newton's laws, but as mass fractions. My justification of this view is available publicly, both as a peer-reviewed article (Kowalski et al., 2021) and a reprint manuscript currently under discussion (Kowalski, 2025), and I invite the authors to join that discussion online as well as here in reply. In short, I believe that inertia, which is fundamental to Newton's laws of motion including the definition of drag forces, must be considered when partitioning transport between diffusive and non-diffusive mechanisms.

It seems clear that we disagree—in the authors' words, regarding the definition of uphill and downhill transport—specifically regarding how these directions are defined by whether molar- or mass-based "concentration" gradients orient diffusion. Here my intention is not to oppose the publication of opposing viewpoints, but rather to stimulate further open discussion. Therefore, I recommend "major revision", and pose *five questions* that I ask the authors to address, at least in reply and at best within a revised version of their manuscript. Some of these questions require building up a background context in which to frame them, and I beg the authors' indulgence in this regard.

***Question 1***: The authors use "mass" in many instances where I think it either undermines the molar-based description of diffusion, or contradicts their own definitions. For example:

> At line 38, they seem to distinguish diffusive and non-diffusive transport, the latter equated with "mass flow" due to drag exerted by exiting water vapour (I agree);

> At line 106, however, equation (5) defines the reference velocity ($u$) based on molar exchanges (I disagree). To simplify, in the static situation where their $u = 0$, the case of opposing/offsetting molar fluxes of water vapour (18 g mol$^{-1}$) and dry air (29 g mol$^{-1}$) does not exclude but rather ensures mass flow. Thus, their definition of diffusion is not independent of mass flow, but necessarily includes it;

> At line 118, the authors note that the total diffusive flux of the mixture is zero, but this is defined on a molar (and not a mass) basis;

> At line 230, the authors again appear to make a clear distinction between "diffusive and mass flows", as if these were independent within their molar framework.

***Would the authors like to revise their manuscript at lines 38 and 230 (see also lines 68, 99, 101, 116, 321, and 371) to refer to "molar flow" rather than "mass flow"?***

***Question 2***: Similarly, the authors identify "mass balance" as one of the three key elements used in their study (line 67), and begin their derivations with equation (1) that is alleged to represent mass balance, despite having molar units. If this seems trivial, it should be noted that the time derivative of their equation (2) yields

$$\frac{dc_t}{dt} = \sum_{\alpha=1}^{n} \frac{dc_\alpha}{dt}$$

which need not be zero since the number of moles is not conserved through certain chemical reactions (see question 4 below), unlike total mass. ***Would the authors like to revise their manuscript at lines 68, 89, 90, 127, 153, and 326 (perhaps 131 and 405 as well) to refer to "molar balance" instead of "mass balance"?***

***Question 3***: Regarding drag forces, I think an analogy is worth examining. Suppose a steel ball of 4cm diameter is in equilibrium at rest on a frictionless horizontal plane. Now, if it experiences a drag force caused by collisions with a swarm of balls moving initially with speed U all in the same direction, perhaps we can distinguish two cases:

A. The moving balls that collide with the steel ball are ping-pong balls of negligible mass (or inertia)—and hence momentum—and therefore hardly drag it along even as they lose much of their original momentum; or

B. The moving balls are also 4cm-diameter steel balls, and if there are many of them the initially static ball ultimately joins the group (whose overall velocity is slightly slowed by its initial inertia—since it began as static—conserving total momentum).

***Is inertia relevant to the drag forces caused by collisions, and does this apply to large collections of molecules as in fluids?***

*Question 4*: Let us consider an initially static system with no external forces applied, consisting of nitrogen dioxide ($NO_2$) on the right ($x > 0$) and ethene (ethylene; $C_2H_4$) on the left ($x < 0$), diffusing in opposite directions with equal flux magnitudes. Since these gases have equal molecular masses (44 g mol$^{-1}$), at first glimpse the distinction between molar-based and mass-based definitions of diffusion seems moot; the initial reference velocity is zero in either case. But if dimerization of $NO_2$ to form dinitrogen tetroxide ($N_2O_4$)

$$2NO_2(g) \rightarrow N_2O_4(g)$$

reduces the number of moles of leftward-diffusing molecules, this proceeding reaction makes the molar-based reference velocity ($u$) that the authors define in their equation (5) increasingly more rightward ($\frac{du}{dt} > 0$). In their molar-based framework, the authors seem to suggest that the time-derivative of $u$ is governed by the Navier-Stokes equation (line 411). ***Does this mean that the system can accelerate with no external forces acting?***

*Question 5*: The answer to Question 4 conditions the follow-up question, which I will therefore specify doubly:

   **2A: *If no, then do the well-established Stefan-Maxwell equations provide a rigorous physical framework for the analysis of transport mechanisms?***

   **2B: *If yes, then do Newton's laws require revision so as not to conflict with the Stefan-Maxwell equations?***

To my mind these questions are revelatory, and I hope the authors will consider them and reply. I furthermore would like to hear the authors' opinions regarding the Kowalski et al. (2021) and Kowalski (2025) analyses that similarly criticise the molar-based framework for representing diffusion as infringing upon Newton's laws. Together with this review, these amount to three documents (ranging from peer-reviewed and published to "grey literature") that provide examples where it seems to me that the molar-based, Stefan-Maxell framework contravenes Newton's laws in defining non-diffusive transport, and therefore in erroneously partitioning net transport and misrepresenting diffusion.

Finally, in case the authors find them to be of use when revising their manuscript, I add the following two specific observations.

**Specific comments**

1.  Beginning at line 324, the authors state that "Although previous studies have recognized that transport of $O_2$ … could be relevant across scales from the canopy to the atmospheric boundary layer (Kowalski et al., 2021), our analysis of the fundamental mass balance equation demonstrates that the primary effects of Stefan flow emerge at the stomatal scale (see Appendix C)." This misses the point that the Kowalski et al. (2021) paper is principally about *turbulent* diffusion, as its title indicates. Its analysis included within the definition of "diffusion" reordering of mass by both molecular motions and turbulence. Appendix C, which considers only molecular diffusion, defines length and velocity scales that are not relevant to the case of atmospheric boundary layer considered by Kowalski et al. (2021), except within millimeters of the surface.
2.  Line 409. Since the molar fraction is dimensionless, equation (C1) has units of $s^{-1}$.

**References**

Kowalski, A. S., Serrano-Ortiz, P., Miranda-Garc a, G., and Fratini, G., "Disentangling turbulent gas diffusion from non-diffusive transport in the boundary layer." *Boundary-Layer Meteorology*, **179** (3), 347-367. https://doi.org/10.1007/s10546-021-00605-5, 2021.

Kowalski, A. S.: Comment on "Technical note: An assessment of the relative contribution of the Soret effect to open water evaporation" by Roderick and Shakespeare (2025), EGUsphere [preprint], https://doi.org/10.5194/egusphere-2025-2814, 2025.

---

## Author Comment (AC1)

A review of "Technical Note: New insights into stomatal oxygen transport viewed as a multicomponent diffusion process" by Vila-Guerau de Arellano et al. (egusphere-2025- 2705)

A. S. Kowalski

**Synopsis**

This study considers the drag exerted by water vapour molecules escaping stomata due to transpiration—the dominant gas exchange process—on oxygen molecules exported from stomata due to photosynthetic production. The issue at hand is the degree to which oxygen transport occurs down its concentration gradient due to Fickian diffusion, versus transport that may be a consequence of mass flow. The study is based on the broadly accepted theory of the Stefan-Maxwell equations, taken to represent the fundamental physics of multicomponent diffusion, and from which a multitude of equations are derived. The authors' mathematical manipulations appear to be correct; I hardly criticize them (but see Specific comments below). Rather, my doubts regarding these analyses concern their starting point, and specifically whether the Stefan-Maxwell equations do indeed accurately represent the fundamental physics of transport mechanisms.

We thank Dr. Kowalski for the time of reading and commenting the paper. In our response, we will limit our response to the scope of the Technical Note. In this context—the transport of oxygen between the substomatal cavity and a stagnant atmosphere at low Reynolds numbers (roughly speaking $10^4$-$10^5$)—we base our theoretical analysis on the Stefan–Maxwell (SM) equations, which, as the referee points out, are broadly accepted in the field. As noted in our reply to referee #2, our starting premise is that the SM equations provide both a rigorous mathematical framework and a physically interpretable description of transport between the substomatal cavity and the atmosphere for the four species considered. This framework is broadly accepted in the field (see Jarman, 1974; von Caemmerer and Farquhar, 1981, and subsequent studies have successfully applied the SM framework for the ternary system; see our Appendix D).

It is important to stress that the diagnostic relations derived from our framework can be coupled to the governing equations for $O_2$, thereby enabling representations across all relevant atmospheric scales, as illustrated in Appendix C. We appreciate the referee's acknowledgement of the correctness of our mathematical formulation. We also recognize that the molar- versus mass-based formulation of diffusion is an important topic in the broader literature, and we will address the referee's comments on this matter in the following points.

**General assessment**

I neither endorse nor oppose publication of this manuscript. I have expressed my assessment that the concentration gradients determining diffusion should be specified, not as molar fractions (as in the Stefan-Maxwell equations) which seem to violate Newton's laws, but as mass fractions. My justification of this view is available publicly, both as a peer-reviewed article (Kowalski et al., 2021) and a reprint manuscript currently under discussion (Kowalski, 2025), and I invite the authors to join that discussion online as well as here in reply. In short, I believe that inertia, which is fundamental to Newton's laws of motion including the definition

of drag forces, must be considered when partitioning transport between diffusive and non-diffusive mechanisms.

In our Technical Note, our focus is limited to the transport of oxygen between the substomatal cavity and a stagnant atmosphere, for which we have adopted the conventional formulation of the Stefan–Maxwell equations in terms of molar quantities (e.g., Curtiss and Hirschfelder, 1949; Bird et al., 2007). This choice is standard in transport theory and enables a direct link between molecular interactions and measurable gas fluxes. Our conclusions are therefore drawn strictly within this established molar-based SM framework at the molecular scale.

Focusing on the results presented in our paper, we note that our findings are similar with those reported in Kowalski et al. (2024), as discussed in our manuscript (see Eqs. 29 and 30). This agreement indicates that, despite differences in formulation between Kowalski et al. (2024) and our study, there is convergence in the predicted relationship between internal $O_2$ mole fractions and atmospheric $O_2$ mole fractions. We prefer to keep our response within the scope of the Technical Note and do not intend to enter a broader debate beyond the specific context of this study. We will further reply to the point raised on mass fractions versus mole fractions in the specific question below.

It seems clear that we disagree—in the authors' words, regarding the definition of uphill and downhill transport—specifically regarding how these directions are defined by whether molar- or mass-based "concentration" gradients orient diffusion. Here my intention is not to oppose the publication of opposing viewpoints, but rather to stimulate further open discussion. Therefore, I recommend "major revision", and pose **five questions** that I ask the authors to address, at least in reply and at best within a revised version of their manuscript. Some of these questions require building up a background context in which to frame them, and I beg the authors' indulgence in this regard.

We appreciate that the referee's intention is to stimulate broader discussion, but we do not intend to further extend our study. In this reply we will therefore focus only on addressing the specific questions insofar as they relate directly to the scope and content of our manuscript.

**Question 1**: The authors use "mass" in many instances where I think it either undermines the molar-based description of diffusion, or contradicts their own definitions. For example:

> At line 38, they seem to distinguish diffusive and non-diffusive transport, the latter equated with "mass flow" due to drag exerted by exiting water vapour (I agree);

> At line 106, however, equation (5) defines the reference velocity ($u$) based on molar exchanges (I disagree). To simplify, in the static situation where their $u = 0$, the case of opposing/offsetting molar fluxes of water vapour (18 g mol$^{-1}$) and dry air (29 g mol$^{-1}$) does not exclude but rather ensures mass flow. Thus, their definition of diffusion is not independent of mass flow, but necessarily includes it;

> At line 118, the authors note that the total diffusive flux of the mixture is zero, but this is defined on a molar (and not a mass) basis;

> At line 230, the authors again appear to make a clear distinction between "diffusive and

mass flows", as if these were independent within their molar framework.

*Would the authors like to revise their manuscript at lines 38 and 230 (see also lines 68, 99, 101, 116, 321, and 371) to refer to "molar flow" rather than "mass flow"?*

We agree with the referee's observation and will revise the manuscript to replace all occurrences of "mass flow" with "molar flow" to ensure consistency with the molar-based Stefan–Maxwell framework used throughout our analysis. The same change will be applied to the specific line references indicated by the referee.

*Question 2*: Similarly, the authors identify "mass balance" as one of the three key elements used in their study (line 67), and begin their derivations with equation (1) that is alleged to represent mass balance, despite having molar units. If this seems trivial, it should be noted that the time derivative of their equation (2) yields

$$\frac{dc_t}{dt} = \sum_{\alpha=1}^{n} \frac{dc_\alpha}{dt}$$

which need not be zero since the number of moles is not conserved through certain chemical reactions (see question 4 below), unlike total mass. *Would the authors like to revise their manuscript at lines 68, 89, 90, 127, 153, and 326 (perhaps 131 and 405 as well) to refer to "molar balance" instead of "mass balance"?*

We agree with the referee's observation and will revise the manuscript to replace "mass balance" with "molar balance" in all relevant locations indicated, to be consistent with the molar units used in our formulation.

*Question 3*: Regarding drag forces, I think an analogy is worth examining. Suppose a steel ball of 4cm diameter is in equilibrium at rest on a frictionless horizontal plane. Now, if it experiences a drag force caused by collisions with a swarm of balls moving initially with speed U all in the same direction, perhaps we can distinguish two cases:

  A. The moving balls that collide with the steel ball are ping-pong balls of negligible mass (or inertia)—and hence momentum—and therefore hardly drag it along even as they lose much of their original momentum; or

  B. The moving balls are also 4cm-diameter steel balls, and if there are many of them the initially static ball ultimately joins the group (whose overall velocity is slightly slowed by its initial inertia—since it began as static—conserving total momentum).

*Is inertia relevant to the drag forces caused by collisions, and does this apply to large collections of molecules as in fluids?*

We appreciate the referee's detailed Questions 3, 4, and 5 and the broader physical considerations they raise. We answer these questions together here. In the context of our Technical Note, we restrict our analysis to the transport of $O_2$ between the substomatal cavity and a stagnant atmosphere under low Reynolds number conditions. As shown in our nondimensional analysis of the $O_2$ governing equation (Appendix C), this regime requires a mathematical description of transport processes that includes both the convective-like molar transport and the diffusive transport driven by concentration gradients. We consider the Stefan–Maxwell equations to provide the most suitable framework for this purpose, as they encompass the coupled effects of multicomponent diffusion and molar transport within a consistent and widely used theoretical formulation.

Our results and conclusions are therefore drawn within the premise that the Stefan–Maxwell equations are valid at the spatial and temporal scales of interest in this study. Questions concerning the applicability of alternative formulations, the explicit role of inertia, or the reconciliation of the SM framework with Newton's laws go beyond the scope of the Technical Note. While these are important topics for broader discussion in the literature, our objective here is to present a self-consistent application of the SM equations to the specific problem of stomatal $O_2$ transport under stagnant atmospheric conditions. We also note that our formulation allows a complete description across scales, with the SM framework applicable at the molecular level, and turbulent diffusion and advection becoming dominant at larger atmospheric spatiotemporal scales. In the revised manuscript, we will emphasize the use of the SM equations and clearly define their range of applicability.

**Question 4**: Let us consider an initially static system with no external forces applied, consisting of nitrogen dioxide (NO2) on the right ($x > 0$) and ethene (ethylene; C2H4) on the left ($x < 0$), diffusing in opposite directions with equal flux magnitudes. Since these gases have equal molecular masses (44 g mol$^{-1}$), at first glimpse the distinction between molar- based and mass-based definitions of diffusion seems moot; the initial reference velocity is zero in either case. But if dimerization of NO2 to form dinitrogen tetroxide (N2O4)

$$2NO_2(g) \rightarrow N_2O_4(g)$$

reduces the number of moles of leftward-diffusing molecules, this proceeding reaction makes the molar-based reference velocity ($u$) that the authors define in their equation (5) increasingly more rightward

$$\left(\frac{du}{dt} > 0\right)$$

In their molar-based framework, the authors seem to suggest that the time-derivative of $u$ is governed by the Navier-Stokes equation (line 411). ***Does this mean that the system can accelerate with no external forces acting?***

***Does this mean that the system can accelerate with no external forces acting?***

**Question 5**: The answer to Question 4 conditions the follow-up question, which I will therefore specify doubly:

   **2A**: ***If no, then do the well-established Stefan-Maxwell equations provide a rigorous***

*physical framework for the analysis of transport mechanisms?*

*2B: If yes, then do Newton's laws require revision so as not to conflict with the Stefan-Maxwell equations?*

To my mind these questions are revelatory, and I hope the authors will consider them and reply. I furthermore would like to hear the authors' opinions regarding the Kowalski et al. (2021) and Kowalski (2025) analyses that similarly criticise the molar-based framework for representing diffusion as infringing upon Newton's laws. Together with this review, these amount to three documents (ranging from peer-reviewed and published to "grey literature") that provide examples where it seems to me that the molar-based, Stefan-Maxell framework contravenes Newton's laws in defining non-diffusive transport, and therefore in erroneously partitioning net transport and misrepresenting diffusion.

We understand that the reviewer is interested in our perspective on Kowalski et al. (2021) and Kowalski (2025); however, we consider this to be outside the scope of both our Technical Note and this reply.

Finally, in case the authors find them to be of use when revising their manuscript, I add the following two specific observations.

**Specific comments**

1.  Beginning at line 324, the authors state that "Although previous studies have recognized that transport of $O_2$ … could be relevant across scales from the canopy to the atmospheric boundary layer (Kowalski et al., 2021), our analysis of the fundamental mass balance equation demonstrates that the primary effects of Stefan flow emerge at the stomatal scale (see Appendix C)." This misses the point that the Kowalski et al. (2021) paper is principally about *turbulent* diffusion, as its title indicates. Its analysis included within the definition of "diffusion" reordering of mass by both molecular motions and turbulence. Appendix C, which considers only molecular diffusion, defines length and velocity scales that are not relevant to the case of atmospheric boundary layer considered by Kowalski et al. (2021), except within millimeters of the surface.

    We thank the referee for this clarification. We will revise the relevant sentence to make clear that Kowalski et al. (2021) focuses principally on turbulent diffusion, with "diffusion" in that work including both molecular motions and turbulence. We will also clarify that our Appendix C considers only molecular diffusion under low Reynolds number conditions. The length and velocity scales derived in the Appendix are relevant only within millimeters of the leaf surface, not to the broader atmospheric boundary layer scales considered in Kowalski et al. (2021). At atmospheric boundary layer scales, temperature and wind differences generate large instabilities, and the flow is organized into coherent turbulent structures. Under such conditions, as shown in our analysis in Appendix C, the Stefan–Maxwell equations are not applicable.

2. Line 409. Since the molar fraction is dimensionless, equation (C1) has units of $s^{-1}$.

We have corrected the units to $s^{-1}$.

**References**

Kowalski, A. S., Serrano-Ortiz, P., Miranda-Garc a, G., and Fratini, G., "Disentangling turbulent gas diffusion from non-diffusive transport in the boundary layer." *Boundary-Layer Meteorology*, **179** (3), 347-367. https://doi.org/10.1007/s10546-021-00605-5, 2021.

Kowalski, A. S.: Comment on "Technical note: An assessment of the relative contribution of the Soret effect to open water evaporation" by Roderick and Shakespeare (2025), EGUsphere [preprint], https://doi.org/10.5194/egusphere-2025-2814, 2024

---

## Author Comment (AC2)

Response to
Reviewer #2
'Mathematically sound, but please consider elaborating on real-world applicability',

The technical note by Vilà-Guerau de Arellano et al. explores to what extent oxygen transport through stomata is regulated by water fluxes in the context of multicomponent mass transfer. The central thesis is that, under low Reynolds number conditions dominated by molecular diffusion, as water fluxes from the evaporating substomatal cavity are typically much higher than oxygen fluxes, the gradient of oxygen across stomata is proportional to water fluxes according to the Stefan–Maxwell equations, leading to uphill diffusion of oxygen that counterbalances the Stefan flow. As the authors put it, this phenomenon "can strongly affect the interpretation of $O_2$ exchange on stomatal level", which I agree on the premise that mass transfer is dominated by multicomponent molecular diffusion and the Stefan flow induced by evaporation. But empirically, it is this premise that I have certain doubts about.

Thanks for reading and commenting on the manuscript. As the referee noted, our starting premise is the use of the Stefan–Maxwell (SM) equations, as they provide both a rigorous mathematical framework and a physically interpretable description of $O_2$ transport between the substomatal cavity and the atmosphere. In this sense, they offer a theoretical basis for interpreting future observational studies as well as a consistent mathematical representation of $O_2$ transport. The resulting expressions are readily applicable for both experimentalists and modelers, as shown by equation (26) and the regime identification presented in Table 2, thereby facilitating the interpretation of future laboratory experiments and their implementation in weather, climate, and carbon–oxygen models that explicitly represent leaf-scale processes

A leaf receiving an intermediate to high level of radiation is typically warmer than the ambient air, because it needs to dissipate heat through diffusion (what meteorologists call sensible heat transfer), in addition to outgoing thermal radiation and latent heat transfer through transpiration. As oxygen and water molecules have different molecular weights, this leaf-to-air temperature gradient may create conditions for thermophoresis in which oxygen molecules move down the temperature gradient (from the substomatal cavity to the air) as opposed to the direction of water vapor thermodiffusion. How thermodiffusion compares to the Stefan flow and molecular diffusion in stomatal oxygen transport seems unknown to me, and it bears consequences for how we interpret leaf-level oxygen exchange measurements.

The referee raises a very interesting and valid point that can serve as a natural follow-up to our study. In meteorological terms, our current analysis assumes neutral stability which means that instabilities due to temperature differences at molecular or turbulent scales are omitted. Including thermal (buoyant) effects, such as temperature gradients between the leaf and the surrounding air, would be a logical next step. Our Technical Note focuses solely on the effects of transport driven by velocity differences arising from the molar flux of water vapor without accounting for these temperature differences. As the referee notes, temperature differences between the atmosphere near the leaf and the substomatal cavity could generate thermally driven instabilities that influence $O_2$ transport.
In the revised manuscript and in the discussion section, we will address the thermodiffusion effect in the Discussion section, citing relevant sources such as Curtiss and Hirschfelder (1949), *Transport Properties of Multicomponent Gas Mixtures* (*Journal of Chemical Physics*, 17), as a potential extension of our neutral-condition theoretical framework. We are also aware of the study by Griffani et al., New Phytologist, 2024 on the effect of the thermodiffusion to transpiration and a similar kind of analysis can be carried out for O2, but we see that it is out of the scope of the present stud

Regarding the interpretation of Stefan–Maxwell diffusion as "the drag forces exerted by all other species," I concur with my fellow reviewer that the underlying physical picture seems murky. It is clear that the authors are alluding to the kinetic theory of gases. But if Stefan–Maxwell diffusion originates from drag forces at the molecular level, in what sense does it differ from viscosity? It seems that this physical picture (lines 36–54) needs to be clarified to build a robust intuition.

Following the advice of the referee, we will revise this part of the Introduction to better place our research in context. In the revised manuscript, we will specify that "drag" (friction) in the Stefan–Maxwell framework refers to momentum exchange in binary molecular collisions between different species, rather than to shear stress in bulk flow, and its relation with viscosity as expressed by the SM diffusion coefficients. In our case, molecular friction arising from differences in molecular velocities—driven by the molar density flux—gives rise to mole gradients. These gradients, in turn, drive diffusive transport at rates determined by the corresponding diffusion coefficients.

I consider the nondimensional treatment in Appendix C a helpful framework for assessing under which conditions Stefan flow and multicomponent molecular diffusion matter. But it seems that the wind speed range quoted in line 425 is the condition in a greenhouse. Wind experienced by top-canopy leaves in a forest can be quite different. It would help to give a threshold of wind speed at which turbulent diffusion becomes more important than multicomponent molecular diffusion.

In real canopies—particularly at the top of the canopy and away from the leaf surface—the flow regime is typically turbulent, and turbulent transport dominates the exchange of oxygen. In our view, the Stefan–Maxwell equations are valid and applicable only under low Reynolds number conditions (below about $10^3$–$10^4$), which occur very close to the leaf when wind speeds approach 0 m s$^{-1}$. As a representative example, for large leaves with a characteristic length scale of ~10 cm, the Reynolds number would be below ~$10^3$, still below the critical value for transition to turbulence. The precise threshold is difficult to determine, as it depends on leaf morphology, flow speed, atmospheric stability, and micro-scale turbulence. The wind speeds given in the manuscript (0.05–0.15 m s$^{-1}$) are intended as illustrative values, representative of potential laboratory or controlled-environment experiments.
In the revised manuscript (Discussion section and in commenting Appendix C), we will clarify the applicability of the SM equations in relation to wind speed values and, following the referee's advice, place our study in a clearer real-world context near the leaf surface. Importantly, at the canopy scale (e.g. eddy covariance flux observations), the effects of Stefan flow are negligible; thus, the real-world applicability of our theory lies primarily at the small, near-leaf scale (see Appendix C).

Lastly, Table 1 presents calculations of molar fluxes of water vapor, $O_2$, and $CO_2$ and their partition into Stefan flow and diffusive flux components. The calculations assume a water vapor mole fraction of 0.01, but in reality, it is the most variable component in the canopy air. In a desert, this value could be much smaller, whereas in a tropical rainforest at 35°C, the air could hold 5.5% water vapor (in mole fractions) at saturation. It would be helpful to expand this table (maybe into a figure) to show calculations under a range of realistic water vapor mole fractions.

Our analysis shows that the most relevant variable is the molar flux density of water, as expressed in Eq. (22). In this equation, the water vapor mole fraction $x_w$ is included in the water vapor gradient (Eq. 22a) but does not appear in the expression for the $O_2$ gradient. Our representative value $x_w$ =0.01 mol mol$^{-1}$ is already relatively high, corresponding to a specific humidity of 16 g$_w$ kg$_a^{-1}$. Observed and simulated values studied in the Amazon rainforest (see Fig. 10 in Pedruzo-Bagazgoitia et al., 2023, *Journal of Advances in Modeling Earth Systems*, 15, e2022MS003210, https://doi.org/10.1029/2022MS003210) range between 20 and 25 g$_w$kg$_a^{-1}$, corresponding to mole

fractions of 0.012–0.016 mol mol$^{-1}$. Below we present a sensitivity analysis with respect to different values of the water vapor mole fraction.

Following the referee's recommendation, we have carried out a sensitivity analysis on $x_w$ over the range 0.01 mol mol$^{-1}$ (very dry conditions, 16 $g_w kg_a^{-1}$ to 0.016 mol mol$^{-1}$ (tropical Amazonia conditions, 25 $g_w kg_a^{-1}$. For the water vapor gradient, we obtained values of –9.45 m$^{-1}$ (dry) and –9.31 (mol mol$^{-1}$) m$^{-1}$ (tropical), compared to the value in Table 1 for $x_w$ = 0.01 (–9.37 (mol mol$^{-1}$) m$^{-1}$). The effect is not very important on the gradients. In the revised manuscript, we will retain the table as it stands but include these additional values in the text to provide a clearer sensitivity analysis of our findings.